# Mesh-Informed Neural Operator : A Transformer Generative Approach

**Yaozhong Shi**                                                                                    *yshi5@caltech.edu*
*California Institute of Technology*

**Zachary E. Ross**                                                                                *zross@caltech.edu*
*California Institute of Technology*

**Domniki Asimaki**                                                                            *domniki@caltech.edu*
*California Institute of Technology*

**Kamyar Azizzadenesheli**                                                                *kaazizzad@gmail.com*
*NVIDIA Corporation*

**Reviewed on OpenReview:** *https://openreview.net/forum?id=K8qAuRfvOG*

## Abstract

Generative models in function spaces, situated at the intersection of generative modeling and operator learning, are attracting increasing attention due to their immense potential in diverse scientific and engineering applications. While functional generative models are theoretically domain- and discretization-agnostic, current implementations heavily rely on the Fourier Neural Operator (FNO), limiting their applicability to regular grids and rectangular domains. To overcome these critical limitations, we introduce the Mesh-Informed Neural Operator (MINO). By leveraging graph neural operators and cross-attention mechanisms, MINO offers a principled, domain- and discretization-agnostic backbone for generative modeling in function spaces. This advancement significantly expands the scope of such models to more diverse applications in generative, inverse, and regression tasks. Furthermore, MINO provides a unified perspective on integrating neural operators with general advanced deep learning architectures. Finally, we introduce a suite of standardized evaluation metrics that enable objective comparison of functional generative models, addressing another critical gap in the field.

## 1 Introduction

Generative models are powerful tools for fields dealing with complex data distributions, with recent advances in diffusion and flow matching models demonstrating impressive capabilities for synthesizing high-fidelity images (Song et al., 2021; Ho et al., 2020; Lipman et al., 2023), audio (Liu et al., 2023; Huang et al., 2023), and video (Jin et al., 2025). These models excel at learning highly complicated probability distributions in finite-dimensional spaces. However, numerous fields in science and engineering–such as seismology, biomechanics, astrophysics and atmospheric sciences—primarily deal with data that inherently reside in infinite-dimensional function spaces. Moreover, in many cases, the data collected in these fields are functions sampled on heterogeneous networks of sensors, or even on manifolds (e.g., global seismic networks or weather stations on Earth's surface). Functional generative models are especially important for these fields because they routinely deal with several broad (sometimes overlapping) challenges: (i) uncertainty quantification in physical units, (ii) the inherent non-uniqueness present in, e.g. solutions to inverse problems, and (iii) the need to model stochastic or latent fields that are effectively unobservable, which are better handled probabilistically. Together, these factors necessitate a paradigm shift towards generative models that operate directly in function spaces.

Recent studies have generalized various generative paradigms to function spaces (Rahman et al., 2022; Shi et al., 2024a;b; Kerrigan et al., 2023a;b; Lim et al., 2025; Seidman et al., 2023) by leveraging neural operators (Azizzadenesheli et al., 2024; Kovachki et al., 2023). Despite these advancements, two critical bottlenecks hinder the broader adoption and rigorous evaluation of functional generative models. First, current implementations are predominantly based on FNO (Li et al., 2021) as their backbone. However, FNO implementations are restricted to regular grids on rectangular domains, thereby preventing the realization of many key theoretical benefits. Second, prior studies rely on dataset-specific metrics, making it difficult to compare functional generative models across datasets and to draw robust conclusions about generative quality.

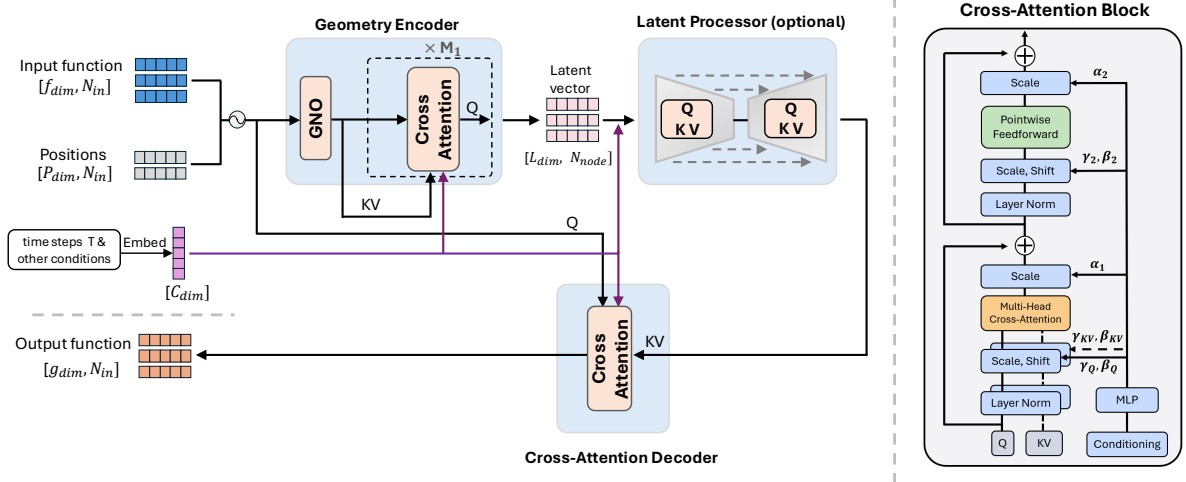

Figure 1: Overview of the MINO architecture. The geometry encoder uses a GNO as a domain-agnostic tokenizer, followed by several cross-attention blocks and an optional latent processor. The decoder then employs a distinct cross-attention mechanism to map the latent representation back to the target locations.

This study addresses the aforementioned limitations of functional generative models. Specifically, our contributions are summarized as follows:

- We introduce MINO, a domain-agnostic functional generative backbone. Our comprehensive experiments show that MINO achieves state-of-the-art (SOTA) performance on a diverse suite of benchmarks with regular and irregular grids.

- We demonstrate through analysis and experiments that Sliced Wasserstein Distance (SWD) and Maximum Mean Discrepancy (MMD) are efficient, robust, dataset-independent metrics for evaluating the performance of functional generative models on regular and irregular grids.

- We propose a novel framework that integrates operator learning with modern deep learning practice. By using a graph neural operator as a domain-agnostic tokenizer and embedding tailored cross-attention modules in both the encoder and decoder, our design avoids information bottlenecks present in prior encoder-processor-decoder neural operators, allowing any powerful, finite-dimensional network (e.g., U-Net or Transformer) to serve as the latent processor.

## 2 Preliminaries

**Neural operators.** Neural operators learn mappings between function spaces, extending deep learning beyond the fixed-dimensional vectors handled by classical networks. This function space perspective is especially important for scientific computing that is governed by partial differential equations (PDEs) (Kovachki et al., 2023; Li et al., 2021). A core theoretical property of neural operators is *discretization convergence (agnosticism)*: as the mesh of input function is refined, the prediction approaches the unique continuous solution (Kovachki et al., 2023). The Fourier Neural Operator (FNO) (Li et al., 2021) achieves this property via spectral-domain convolutions, achieving quasi-linear complexity on regular grids. Its global mixing excels

at capturing long-range dependencies but may miss fine-scale detail (required for generative tasks) unless many Fourier modes are retained (Liu-Schiaffini et al., 2024).

For handling functions on irregular grids, Graph Neural Operator (GNO) (Gilmer et al., 2017; Li et al., 2020) remains a powerful candidate. GNO shares the message-passing functionality of Graph Neural Network (GNN), but differs in that the search radius for neighbors is defined in physical coordinates and made consistent across resolutions to guarantee discretization convergence. This results in GNO having a local receptive field, however it is computationally inefficient for certain tasks (Li et al., 2023; Liu-Schiaffini et al., 2024) and often struggles to learn the high-frequency information required for many generative models. This difficulty arises from the standard message-passing mechanisms, which tend to act as low-pass filters (Li et al., 2018; Giovanni et al., 2023). More recent work has focused on improved scalability and performance aspects of Neural Operators on irregular grids and non-rectangular physical domains, e.g., Geometry-Informed Neural Operator (GINO), Universal Physical Transformer (UPT), Latent Neural Operator (LNO), and Transolver (Li et al., 2023; Alkin et al., 2024; Wang & Wang, 2024; Wu et al., 2024). However, it remains unclear whether these architectures will also be effective for functional generative modeling on irregular grids (see Appendix C).

**Functional generative models.** Classical generative models learn a mapping from a simple base distribution (e.g., multi-variate Gaussian) to a target finite dimensional data distribution. Alternatively, the functional perspective frames this problem in infinite-dimensional spaces. A sample function $f_1 : D \to \mathbb{R}^{f_{dim}}$, with $D$ denoting the domain of the function, is treated as a sample from a target probability measure $\mu_1$ over a function space. The discrete samples are formed by evaluating $f_1$ on a set of grid points of locations. Functional generative models often aim to learn a mapping between function spaces by training a neural operator to transform a function $f_0$ sampled from a simple base measure $\mu_0$ (typically a Gaussian random field measure $\mathcal{N}(0, C)$) into a function $f_1$ that follows the target measure $\mu_1$. The distribution of the discrete data is thus a finite-dimensional marginal of the true data measure $\mu_1$.

Adopting this perspective, functional generative models offer several key benefits not generally available in their traditional finite-dimensional counterparts:

- *Flexible domain geometries:* The function domain $D$ is not required to be rectangular; it could be an irregular domain or even a manifold. Such flexibility is ideal for many real-world scenarios in science and engineering, such as modeling rainfall over a city with an irregular boundary, or modeling weather patterns on the Earth's surface.

- *Discretization agnosticism (mesh invariance):* By learning a probability measure over functions, functional generative models can be trained on samples with varying discretizations and can subsequently generate new function samples at any desired discretization in a zero-shot manner.

- *Inference-time control and guidance in function space:* Functional diffusion/flow models enable the incorporation of inference-time scaling rules and precise guidance signals directly within function spaces. For instance, this facilitates the development of probabilistic PDE solvers that can rigorously enforce hard boundary conditions or other external controls during generation (Cheng et al., 2024; Yao et al., 2025).

- *Universal functional regression via stochastic process learning:* By leveraging invertible trajectories derived from neural operators, it is possible to perform functional regression with learned (non-Gaussian) stochastic process priors (Shi et al., 2024a; 2025). These learned distributions are capable of providing exact prior and posterior density estimation for a general stochastic process.

**Flow matching in Hilbert space.** Recent work has established a rigorous mathematical framework for extending flow matching to Hilbert spaces (Kerrigan et al., 2023b), Functional Flow Matching (FFM), where a velocity field is learned to transport a base Gaussian measure to a target measure. This was further built upon by Operator Flow Matching (OFM) (Shi et al., 2025), which extended the paradigm to stochastic processes and incorporated a dynamic optimal-transport path.

The core idea of FFM/OFM is to learn a continuous path of functions, $f_t$ for $t \in [0, 1]$, that transforms samples $f_0$ from a base Gaussian measure $\mu_0$ into samples $f_1$ from a target data measure $\mu_1$. This transformation is

governed by an Ordinary Differential Equation (ODE) whose velocity field, $v_\theta$, is parameterized by a neural operator with weights $\theta$:

$$\frac{df_t}{dt} = v_\theta(f_t, t) \tag{1}$$

For training, OFM draws pairs of functions $(f_0, f_1)$ from a mini-batch optimal coupling $\pi(\mu_0, \mu_1)$, which is achieved by minimizing the 2-Wasserstein distance between $\mu_0$ and $\mu_1$. The base measure, $\mu_0$, is typically a Gaussian measure $\mathcal{N}(0, C)$, where $C$ is a trace-class covariance operator. The framework then defines the path $f_t$ as a linear interpolation between $f_0$ and $f_1$, i.e., $f_t = (1-t)f_0 + tf_1$. $f_t$ induces a time-dependent probability measure $\mu_t$, such that at $t = 0, \mu_t = \mu_0$ and $t = 1, \mu_t = \mu_1$. This specific choice yields a simple ground truth expression for the velocity, $v_t = f_1 - f_0$. The training objective is then to minimize the Mean Squared Error (MSE) between the parameterized velocity field $v_\theta(f_t, t)$ and this target $v_t$. To generate a new sample, one draws $f_0 \sim \mu_0$ and solves the learned ODE numerically. While functional diffusion models offer a related framework, they typically rely on Stochastic Differential Equations (SDEs), which introduce an additional perturbation term with longer generation time and inferior performance compared to ODE-based flow matching (Lim et al., 2025; Kerrigan et al., 2023a). A fundamental property of both paradigms is *domain alignment*: for a given trajectory, all evolving functions $f_t$ must be defined on the same fixed spatial domain $D$. ($\mu_t \in \mathcal{P}(L^2(D; \mathbb{R}^{f_{dim}}))$). Crucially, the theory allows this domain $D$ to be geometrically complex; it is not restricted to a rectangular shape and can be a domain with an irregular boundary or even a manifold. However, this theoretical flexibility has been underutilized in practice, as existing implementations predominantly rely on FNO, which restricts them to domains discretized by regular grids.

**Grid-Based Architectures.** The modern U-Net architecture used in diffusion models (Rombach et al., 2022) has become a de-facto standard for generative tasks on grid-like data. These advanced U-Nets are distinct from the variant used in some earlier neural operators (Rahman et al., 2023), the U-Net can effectively capture multi-scale information through their hierarchical structure, skip connections, and integration of attention and residual blocks. Nevertheless, they remain constrained to regular grids due to their reliance on standard convolutional neural network kernels. Another prominent line of work involves Transformer-based architectures. Standard Vision Transformers (ViTs) and Diffusion Transformers (DiTs) (Dosovitskiy et al., 2021; Peebles & Xie, 2023), first "patchify" the input into non-overlapping blocks and then apply full self-attention, incurring $O(N_{\text{patch}}^2)$ time complexity. For images, $N_{\text{patch}} \ll H \times W$ (Height and Width), yet the patchify operation creates discontinuities at block borders that can lead to checkerboard artifacts (Fang et al., 2023). Moreover, both ViT and DiT presuppose a fixed rectangular grid so that patchification is well defined.

Table 1: Comparison of MINO with other architectures. "local $\to$ global" receptive field gathers global context through stacked local operations. "local + global" receptive field combines local and global information directly. A detailed discussion is provided in Appendix C

| Architecture | Efficient | Receptive field | Irregular grid | Discretization convergent | Frequency learnt |
|---|---|---|---|---|---|
| GNN | ✗ | local | ✓ | ✗ | low |
| GNO | ✗ | local | ✓ | ✓ | low |
| FNO | ✓ | global | ✗ | ✓ | low + high |
| U-Net | ✓ | local $\to$ global | ✗ | ✗ | low + high |
| ViT/DiT | ✓ | global | ✗ | ✗ | low + high |
| GINO | ✓ | local $\to$ global | ✓ | ✓ | low |
| UPT | ✓ | local $\to$ global | ✓ | ✓ | low |
| **MINO [Ours]** | ✓ | local + global | ✓ | ✓ | low + high |

## 3 Methods

Our proposed architecture, MINO, provides an effective solution for functional generative models on non-rectangular domains with irregular grids. It integrates powerful deep learning backbones with neural operators in a manner that is tailored for improving functional generative performance. Specifically, this is achieved

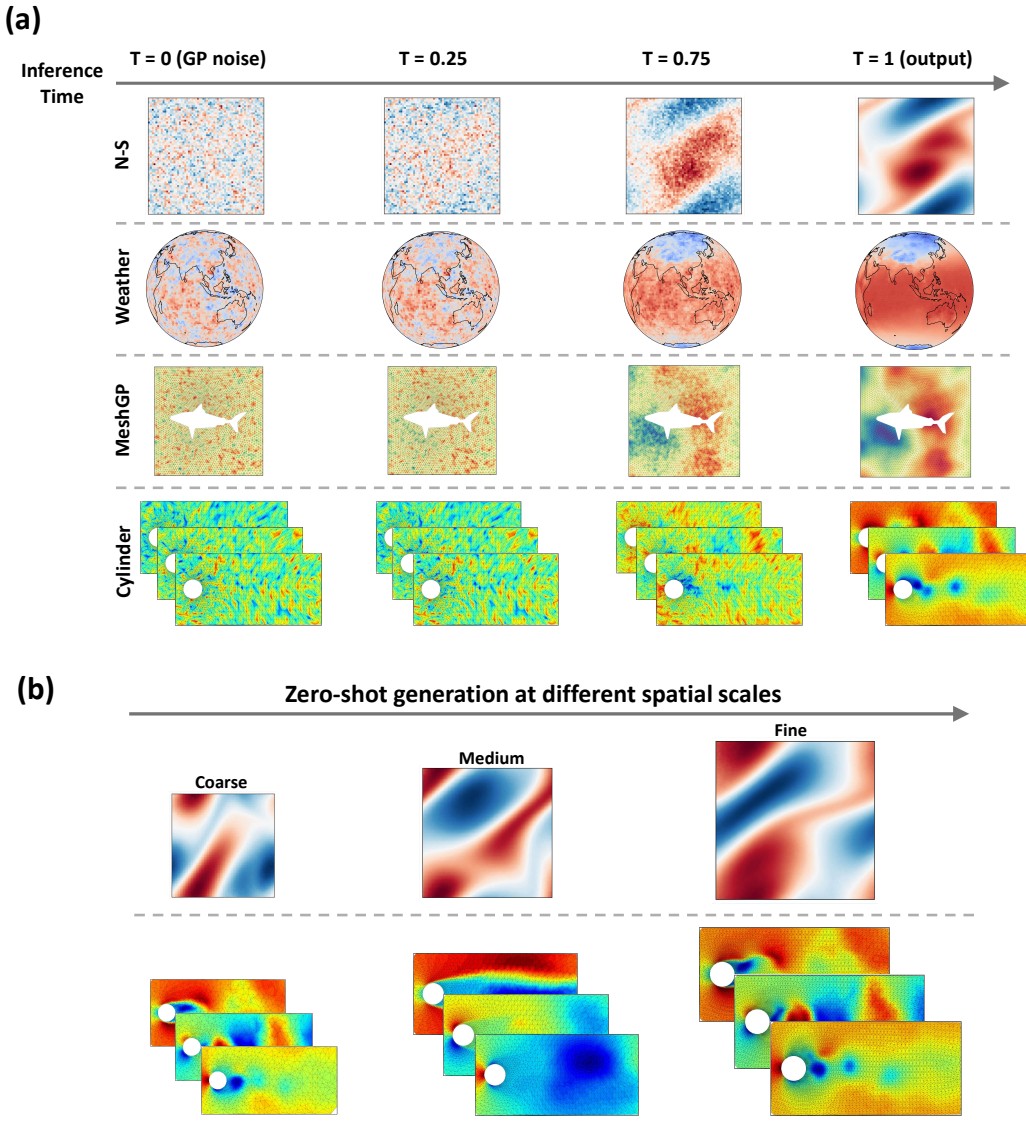

Figure 2: Inference and zero-shot Generation with MINO. (a) MINO gradually transforms a GP sample to the data sample under flow matching paradigm. (b) Zero-shot generation at varying spatial scales by directly transforming finer GP samples to finer data samples.

by combining GNO with cross-attention mechanisms (see Table 1). As illustrated in Figure 1, the GNO in the encoder can be viewed as a domain- and discretization-agnostic alternative to the "patchify" operation. By injecting the input function representation into both the encoder and the cross-attention decoder, our framework benefits from both local and global receptive fields.

In the following, we outline the MINO architecture. We specifically focus on its use in the flow matching paradigm to learn velocity fields, as illustrated in Fig. 2. Although our primary application is generative modeling, the MINO architecture is general-purpose and can be used for other operator learning tasks, such as solving PDEs, with minor adjustments.

**Problem Formulation.** Within the flow matching paradigm, MINO takes three primary inputs: a noisy function $\mathbf{f_t}$, the set of its observation positions $\mathbf{f_{pos}}$, and the corresponding time step $\mathbf{t}$, along with any optional conditioning variables. Specifically, the input function $\mathbf{f_t}$ is represented by its values at $N_{\text{in}}$ discrete

locations, $\mathbf{f_{pos}} = \{p_i\}_{i=1}^{N_{in}}$. These locations are a discretization of a continuous domain $D \subset \mathbb{R}^{P_{dim}}$. Note that $N_{in}$ can be different per sample. The codomain (channel) of $\mathbf{f}_t$ has dimension $f_{dim}$, resulting in an input tensor of function values with shape $[f_{dim}, N_{in}]$. The output is another function (velocity field $\mathbf{v}_t$) at these same locations. In the context of flow matching or diffusion, the output dimension matches the input, so the target output also has a shape of $[g_{dim}, N_{in}]$, where $g_{dim} = f_{dim}$.

The MINO framework is inspired by the encoder-processor-decoder structure of prior architectures like GINO and UPT. Our model consists of three main components: (i) a geometry encoder, (ii) an (optional) latent-space processor, and (iii) a cross-attention decoder. We introduce several key modifications to this design that significantly improve performance on functional generative tasks, as detailed in the following subsections.

**Geometry Encoder.** First, we apply sinusoidal embeddings (Vaswani et al., 2017; Peebles & Xie, 2023) to the time step $\mathbf{t}$ and the observation positions $\mathbf{f_{pos}}$, yielding $\mathbf{t_{emb}}$ and $\mathbf{p_{emb}}$ respectively, as shown in Eq. 2. The input function $\mathbf{f}_t$ is then concatenated with the embedded positions $\mathbf{p_{emb}}$ along the codomain before being passed to a GNO layer; this maps the function from its potentially irregular discretization $\{p_i\}_{i=1}^{N_{in}}$ to a latent representation on a predefined regular grid of query points $\{p_j^{query}\}_{j=1}^{N_{node}}$. The GNO mapping is based on neighbor-searching within a fixed radius $r$, where $N_{node} \ll N_{in}$. Following this GNO layer, a linear layer projects its output $\mathbf{f_t^E}$ into $\mathbf{h_0^E}$ of shape $[L_{dim}, N_{node}]$.

This initial latent representation $\mathbf{h_0^E}$ is then processed by $M_1$ blocks of Multi-Head Cross-Attention (MHCA). Importantly, the key-value ($KV$) pair for all attention blocks is fixed as $\mathbf{h_0^E}$, while the query ($Q$) is the output of the previous MHCA layer $\mathbf{h_{j-1}^E}$, and the time embedding $\mathbf{t_{emb}}$ is used for conditioning. We found this configuration yields better performance than the standard self-attention used in DiT, with comparisons provided in Appendix E. Thus, for this entire component of the architecture, we have the following,

$$\mathbf{p_{emb}} = \text{Emb}(\mathbf{f_{pos}}), \quad \mathbf{t_{emb}} = \text{Emb}(\mathbf{t}) \tag{2}$$

$$\mathbf{f_t^E} = \text{GNO}(\text{concat}(\text{MLP}(\mathbf{f_t}), \mathbf{p_{emb}})), \quad \mathbf{h_0^E} = \text{Linear}(\mathbf{f_t^E}) \tag{3}$$

$$\mathbf{h_j^E} = \text{MHCA}(Q = \mathbf{h_{j-1}^E}, KV = \mathbf{h_0^E}, C = \mathbf{t_{emb}}), \qquad j = 1, \cdots, M_1. \tag{4}$$

The output of the geometry encoder has a fixed size of $[L_{dim}, N_{node}]$, which is independent of the manner in which the input function $\mathbf{f_t}$ is discretized.

**Latent-Space Processor.** Given that the latent representation $\mathbf{h_{M_1}^E}$ has a fixed size, any suitable neural network (NN) can act as the processor. In our implementation, we use a Diffusion U-Net, known for being a strong backbone in generative modeling. It is important to note that this processor is optional; MINO remains a well-defined neural operator without it, although we found that its inclusion generally improves performance. The processor's input is conditioned on $\mathbf{t_{emb}}$, whereas its output, $\mathbf{h^P}$, maintains the same shape as its input,

$$\mathbf{h^P} = \text{NN}(\mathbf{h_{M_1}^E}, \mathbf{t_{emb}}) \tag{5}$$

**Cross-Attention Decoder.** The decoder employs a different cross-attention mechanism from that of the encoder to enable mapping the latent representation back into the output velocity field at the original observation locations. Specifically, $\mathbf{f_t}$ is first processed by an MLP and then concatenated with $\mathbf{p_{emb}}$ to form $\mathbf{f_t^D}$. In a single cross-attention block, $\mathbf{f_t^D}$ serves as the query ($Q$), while the key-value ($KV$) pair is taken from the output of the optional processor $\mathbf{h^P}$ (or $\mathbf{h_{M_1}^E}$ if the processor is omitted). The time embedding $\mathbf{t_{emb}}$ provides conditioning. Finally, a LayerNorm and a linear transformation are applied to the output of this attention block to produce the estimated velocity field $\mathbf{v_t}$.

$$\mathbf{f_t^D} = \text{MLP}(\text{concat}(\text{MLP}(\mathbf{f_t}), \mathbf{p_{emb}})) \tag{6}$$

$$\mathbf{h_1^D} = \text{MHCA}(Q = \mathbf{f_t^D}, KV = \mathbf{h^P}, C = \mathbf{t_{emb}}) \tag{7}$$

$$\mathbf{v_t} = \text{Linear}(\text{LayerNorm}(\mathbf{h_1^D})) \tag{8}$$

A detailed comparison of the MINO framework to prior neural operator architectures is provided in Appendix C, with an ablation study of our model's components in Appendix E

## 4  Experiments

In this section we empirically evaluate MINO and other baselines on a suite of functional generative benchmarks under the OFM (Shi et al., 2025) paradigm due to its concise formulation and SOTA performance among functional generative paradigms. For each task we parameterize the velocity field $v_\theta(f_t, t)$ with different neural-operator backbones to study how the choice of architecture affects performance. The benchmarks cover both regular and irregular grids, providing a comprehensive assessment. To the best of our knowledge, this is the first systematic comparison of modern neural-operator architectures on function-generation problems. We evaluate two variants of our proposed architecture: (i) MINO-U: a Diffusion U-Net as the latent space processor, and (ii) MINO-T: a pure Transformer-based variant without the latent space processor, where the encoder receives additional cross-attention blocks.

Table 2: Summary of experimental benchmarks, covering diverse domains with both regular and irregular grids

| Geometry | Datasets | Domain | Co-domain | Mesh | Training Samples | Test Samples |
|---|---|---|---|---|---|---|
| Regular Grid | Navier Stokes | $D \subset \mathbb{R}^2$ | 1 | 4,096 | 30,000 | 5,000 |
| | Shallow Water | $D \subset \mathbb{R}^2$ | 1 | 4,096 | 30,000 | 5,000 |
| | Darcy Flow | $D \subset \mathbb{R}^2$ | 1 | 4,096 | 8,000 | 2,000 |
| Irregular Grid | Cylinder Flow | $D \subset \mathbb{R}^2$ | 3 | 1,699 | 30,000 | 5,000 |
| | MeshGP | $D \subset \mathbb{R}^2$ | 1 | 3,727 | 30,000 | 5,000 |
| | Global Climate | $\mathbb{S}^2 \subset \mathbb{R}^3$ | 1 | 4,140 | 9,676 | 2,420 |

**Datasets.** We curated a benchmark of six challenging functional datasets to evaluate performance across different domain types: (i) Rectangular Domains and Regular Grids: We use Navier-Stokes (Li et al., 2021), Shallow Water equation (Takamoto et al., 2022), and Darcy Flow (Takamoto et al., 2022) datasets. (ii) Irregular Domains and Grids: We use the Cylinder Flow dataset (Han et al., 2022), a synthetic Mesh-GP dataset on irregular meshes (Zhao et al., 2022), and a real-world global climate dataset (Dupont et al., 2022). These datasets are summarized in Table 2. Detailed descriptions of the datasets and their associated preparation are provided in Appendix A.

**Baselines.** To ensure a thorough comparison, we selected several state-of-the-art neural operators tailored to different domains : (i) For irregular grids, we benchmark against leading architectures including the Universal Physical Transformer (UPT) (Alkin et al., 2024), Latent Neural Operator (LNO) (Wang & Wang, 2024), Transolver (Wu et al., 2024), and Geometry-Informed Neural Operator (GINO) (Li et al., 2023). (ii) For regular grids, we further include the Fourier Neural Operator (FNO) as a strong, established baseline. To ensure a fair assessment, the parameter counts (or computational budgets) of all baseline models were carefully matched to our variants. It is important to note that, with the exception of FNO, all evaluated models (including our MINO variants) are designed to be domain-agnostic and process regular and irregular grids identically. A detailed description of the the baseline implementation is provided in Appendix A.

**Metrics.** A significant current challenge in the field of functional generative models is the lack of standardized evaluation protocols. Previous studies often rely on dataset-specific metrics, which prohibits fair and direct comparison between models (Rahman et al., 2022; Kerrigan et al., 2023a; Shi et al., 2024a; Lim et al., 2025). To address this gap, our work validates two robust and general metrics suitable for estimating the distance between the learned probability measure $\nu$ (from which generated samples $X$ are drawn) and the target probability measure $\mu$ (from which test samples (dataset) $Y$ are drawn).

The first metric is the Sliced Wasserstein Distance (SWD) (Bonneel et al., 2015; Han, 2023). SWD provides a computationally efficient yet powerful alternative to the exact Wasserstein distance, which is often intractable in high dimensions. It works by projecting high-dimensional distributions onto random 1D lines and then averaging the simpler 1D Wasserstein distances. This approach converges to the true Wasserstein distance as the number of projections increases and provides a robust way to compare the geometric structure of two probability measures. For two measures $\mu$ and $\nu$ in Hilbert space, SWD is defined as (Definition 2.5 of Han

(2023)):

$$\mathrm{SWD}_p^\gamma(\mu, \nu) = \left( \frac{1}{\gamma_S(S)} \int_{\|\theta\|=1} W_p^p(\theta_\# \mu, \theta_\# \nu) \gamma_S(d\theta) \right)^{\frac{1}{p}} \tag{9}$$

Where base measure $\gamma_s$ is strictly positive Borel measure defined on a sphere $S$, $\mathcal{W}_p$ is the Wasserstein-p distance, and $\theta_\# \mu, \theta_\# \nu$ denote the pushforward measure of $\mu, \nu$ under the projection $\theta$ separately. Further details are provided in Appendix B.

The second metric is the Maximum Mean Discrepancy (MMD) (Gretton et al., 2012). Let $\mathcal{X} = L^2(D; \mathbb{R}^{f_{dim}})$ and choose a bounded, characteristic kernel $k : \mathcal{X} \times \mathcal{X} \to \mathbb{R}$ with reproducing-kernel Hilbert space (RKHS) $(\mathcal{H}_k, \langle \cdot, \cdot \rangle_k)$. Denote the mean embeddings $m_\mu = \mathbb{E}[k(X, \cdot)]$, $m_\nu = \mathbb{E}[k(Y, \cdot)]$. Then MMD is defined as

$$\mathrm{MMD}_k(X, Y) = \left\| m_\nu - m_\mu \right\|_{\mathcal{H}_k} \tag{10}$$

Because $k$ is characteristic, $\mathrm{MMD}_k = 0$ if and only if $\mu = \nu$. In Appendix B, we show how to calculate SWD and MMD given discretized function samples, and empirically verify that these metrics are valid, robust, and sample-efficient for functional generative models.

**Results.** A visualization of the performance of MINO is provided in Figure 3 and as shown in Table 4, on regular grid benchmarks, our MINO variants achieve SOTA performance: MINO-U achieves best performance on the Navier-Stokes and Shallow Water datasets, while MINO-T excels on the Darcy Flow benchmark. In contrast, several baseline neural operators (GINO, LNO, UPT) perform poorly. Further analysis in Appendix C, D suggests this is because these models primarily learn low-frequency information and fail to capture the high-frequency components essential for these generative tasks. FNO and Transolver achieve comparable, albeit inferior, results to our models. Notably, beyond superior accuracy, our MINO variants are also significantly more efficient than Transolver. As detailed in Table 5, they require substantially less computation to train, are more GPU memory-efficient, and achieve up to a 3.0x speedup during inference, demonstrating the efficiency of the MINO architecture.

For tasks on irregular grids, our models continue to demonstrate top performance. MINO-U achieves the best results on the Cylinder Flow benchmark, while MINO-T is the top performer on the Mesh-GP and Global Climate benchmarks. Then, We perform an ablation study, which confirm the effectiveness the each component of our architecture, as detailed in Appendix E. Last, we provide additional experiments in Appendix F and G to further demonstrate the superior performance of MINO.

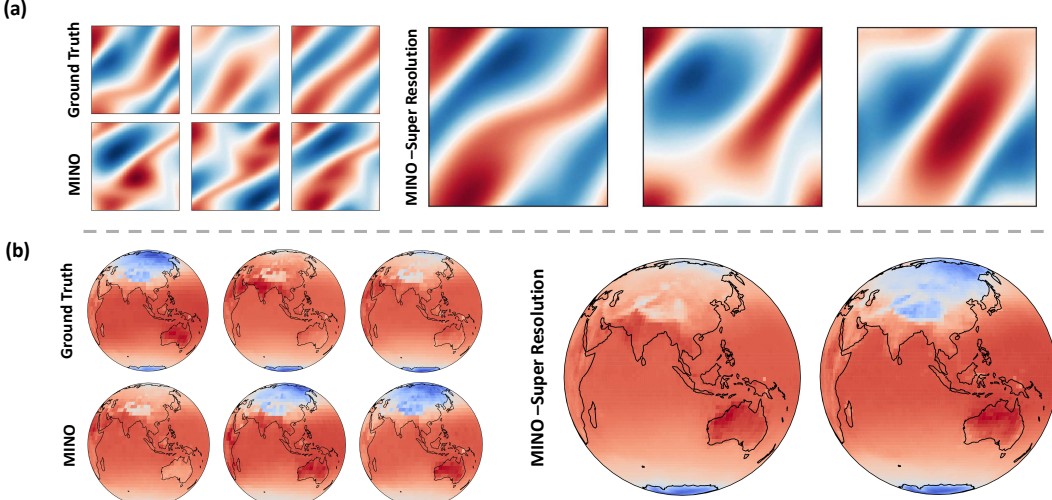

Figure 3: Visualization of generation and zero-shot super-resolution by MINO-U. (a) Navier-Stokes samples generated on the original mesh (4,096 nodes) and a finer mesh (25,600 nodes). (b) Global Climate sample generated on the original mesh (4,140 nodes) and a finer mesh (16,560 nodes).

Table 3: Generation performance on benchmarks with irregular grids. Best performance in bold

| Dataset → | Cylinder Flow | | Mesh-GP | | Global Climate | |
|---|---|---|---|---|---|---|
| Model ↓ Metric → | SWD | MMD | SWD | MMD | SWD | MMD |
| GINO | $4.6 \cdot 10^{-1}$ | $3.1 \cdot 10^{-1}$ | $1.1 \cdot 10^{0}$ | $3.6 \cdot 10^{-1}$ | $6.9 \cdot 10^{-1}$ | $5.1 \cdot 10^{-1}$ |
| UPT | $7.3 \cdot 10^{-1}$ | $5.4 \cdot 10^{-1}$ | $4.2 \cdot 10^{-1}$ | $2.8 \cdot 10^{-1}$ | $7.4 \cdot 10^{-1}$ | $5.4 \cdot 10^{-1}$ |
| Transolver | $2.5 \cdot 10^{-2}$ | $2.4 \cdot 10^{-2}$ | $9.1 \cdot 10^{-2}$ | $4.6 \cdot 10^{-2}$ | $2.8 \cdot 10^{-2}$ | $2.9 \cdot 10^{-2}$ |
| LNO | $5.3 \cdot 10^{-1}$ | $3.8 \cdot 10^{-1}$ | $3.5 \cdot 10^{-1}$ | $2.4 \cdot 10^{-1}$ | $6.7 \cdot 10^{-1}$ | $4.9 \cdot 10^{-1}$ |
| MINO-T (ours) | $2.9 \cdot 10^{-2}$ | $2.6 \cdot 10^{-2}$ | $\mathbf{4.1 \cdot 10^{-2}}$ | $\mathbf{3.0 \cdot 10^{-3}}$ | $\mathbf{2.1 \cdot 10^{-2}}$ | $\mathbf{1.4 \cdot 10^{-2}}$ |
| MINO-U (ours) | $\mathbf{2.3 \cdot 10^{-2}}$ | $\mathbf{2.1 \cdot 10^{-2}}$ | $7.2 \cdot 10^{-2}$ | $4.1 \cdot 10^{-2}$ | $2.8 \cdot 10^{-2}$ | $2.6 \cdot 10^{-2}$ |

Table 4: Generation performance on benchmarks with regular grids. Evaluation metrics include the Sliced Wasserstein Distance (SWD), Maximum Mean Discrepancy (MMD), and Mean Squared Error (MSE) for spectra, autocovariance, and point-wise density. Best performance is indicated in bold.

| Datasets | Model ↓ Metric → | SWD | MMD | Spectra-MSE | Autocovariance-MSE | Density-MSE |
|---|---|---|---|---|---|---|
| Navier-Stokes | GINO | $4.6 \cdot 10^{-1}$ | $3.7 \cdot 10^{-1}$ | $1.9 \cdot 10^{2}$ | $4.6 \cdot 10^{-4}$ | $1.9 \cdot 10^{-2}$ |
| | UPT | $6.0 \cdot 10^{-1}$ | $4.7 \cdot 10^{-1}$ | $2.5 \cdot 10^{3}$ | $1.5 \cdot 10^{-2}$ | $4.0 \cdot 10^{-3}$ |
| | Transolver | $5.3 \cdot 10^{-2}$ | $5.1 \cdot 10^{-2}$ | $1.1 \cdot 10^{2}$ | $6.7 \cdot 10^{-4}$ | $2.0 \cdot 10^{-4}$ |
| | LNO | $5.2 \cdot 10^{-1}$ | $3.8 \cdot 10^{-1}$ | $2.0 \cdot 10^{4}$ | $1.1 \cdot 10^{-1}$ | $5.0 \cdot 10^{-4}$ |
| | FNO | $3.2 \cdot 10^{-2}$ | $2.6 \cdot 10^{-2}$ | $5.1 \cdot 10^{1}$ | $2.2 \cdot 10^{-4}$ | $6.9 \cdot 10^{-5}$ |
| | MINO-T (Ours) | $4.0 \cdot 10^{-2}$ | $3.6 \cdot 10^{-2}$ | $1.9 \cdot 10^{1}$ | $\mathbf{6.8 \cdot 10^{-5}}$ | $1.6 \cdot 10^{-5}$ |
| | MINO-U (Ours) | $\mathbf{2.8 \cdot 10^{-2}}$ | $\mathbf{1.9 \cdot 10^{-2}}$ | $\mathbf{1.3 \cdot 10^{1}}$ | $9.2 \cdot 10^{-5}$ | $\mathbf{1.4 \cdot 10^{-5}}$ |
| Shallow Water | GINO | $7.3 \cdot 10^{-1}$ | $5.1 \cdot 10^{-1}$ | $1.5 \cdot 10^{2}$ | $1.8 \cdot 10^{-1}$ | $9.9 \cdot 10^{-1}$ |
| | UPT | $8.6 \cdot 10^{-1}$ | $5.9 \cdot 10^{-1}$ | $3.1 \cdot 10^{2}$ | $4.0 \cdot 10^{-3}$ | $1.0 \cdot 10^{0}$ |
| | Transolver | $1.7 \cdot 10^{-2}$ | $1.8 \cdot 10^{-2}$ | $2.0 \cdot 10^{-2}$ | $1.2 \cdot 10^{-6}$ | $9.0 \cdot 10^{-4}$ |
| | LNO | $8.7 \cdot 10^{-1}$ | $6.5 \cdot 10^{-1}$ | $2.1 \cdot 10^{2}$ | $1.4 \cdot 10^{0}$ | $1.0 \cdot 10^{0}$ |
| | FNO | $1.0 \cdot 10^{-2}$ | $9.4 \cdot 10^{-3}$ | $7.9 \cdot 10^{-3}$ | $2.8 \cdot 10^{-6}$ | $1.0 \cdot 10^{-4}$ |
| | MINO-T (Ours) | $9.8 \cdot 10^{-3}$ | $8.7 \cdot 10^{-3}$ | $1.6 \cdot 10^{-2}$ | $5.2 \cdot 10^{-6}$ | $2.0 \cdot 10^{-5}$ |
| | MINO-U (Ours) | $\mathbf{4.0 \cdot 10^{-3}}$ | $\mathbf{1.1 \cdot 10^{-3}}$ | $\mathbf{4.3 \cdot 10^{-3}}$ | $\mathbf{5.1 \cdot 10^{-7}}$ | $\mathbf{1.7 \cdot 10^{-5}}$ |
| Darcy Flow | GINO | $4.0 \cdot 10^{-1}$ | $3.0 \cdot 10^{-1}$ | $5.5 \cdot 10^{3}$ | $5.4 \cdot 10^{-2}$ | $1.8 \cdot 10^{-3}$ |
| | UPT | $8.7 \cdot 10^{-1}$ | $4.6 \cdot 10^{-1}$ | $6.0 \cdot 10^{4}$ | $4.0 \cdot 10^{0}$ | $7.5 \cdot 10^{-3}$ |
| | Transolver | $2.2 \cdot 10^{-1}$ | $8.9 \cdot 10^{-2}$ | $4.8 \cdot 10^{3}$ | $3.4 \cdot 10^{-1}$ | $3.0 \cdot 10^{-4}$ |
| | LNO | $5.5 \cdot 10^{-1}$ | $3.8 \cdot 10^{-1}$ | $2.7 \cdot 10^{3}$ | $2.4 \cdot 10^{-1}$ | $1.0 \cdot 10^{-3}$ |
| | FNO | $1.1 \cdot 10^{-1}$ | $3.6 \cdot 10^{-2}$ | $1.2 \cdot 10^{3}$ | $8.4 \cdot 10^{-2}$ | $8.9 \cdot 10^{-5}$ |
| | MINO-T (Ours) | $\mathbf{8.9 \cdot 10^{-2}}$ | $\mathbf{3.4 \cdot 10^{-2}}$ | $\mathbf{9.1 \cdot 10^{2}}$ | $\mathbf{4.4 \cdot 10^{-2}}$ | $9.4 \cdot 10^{-5}$ |
| | MINO-U (Ours) | $9.5 \cdot 10^{-2}$ | $4.8 \cdot 10^{-2}$ | $1.0 \cdot 10^{3}$ | $7.7 \cdot 10^{-2}$ | $\mathbf{3.8 \cdot 10^{-5}}$ |

Table 5: Computational efficiency and performance comparison between MINO variants and Transolver during training and inference on Navier-Stokes benchmark.

| Training Phase→ | Params | GPU memory (per sample) | Training time (per epoch) | Speedup |
|---|---|---|---|---|
| Transolver | 15.0 M | 0.913 GB | 734 s | 1× |
| MINO-T (ours) | 21.5 M | 0.429 GB | **277 s** | **2.6×** |
| MINO-U (ours) | 19.2 M | **0.419 GB** | 279 s | 2.6× |
| Inference Phase → | SWD | MMD | Generation time (per sample) | Speedup |
| Transolver | 0.053 | 0.051 | 1.49 s | 1× |
| MINO-T (ours) | 0.040 | 0.036 | 0.51 s | 2.9× |
| MINO-U (ours) | **0.028** | **0.019** | **0.49 s** | **3.0×** |

# 5 Conclusions

In this paper, we introduce the Mesh-Informed Neural Operator (MINO), a novel backbone for functional generative models that leverages graph neural operators and cross-attention mechanisms. MINO operates

directly on arbitrary meshes, addressing the reliance on grid-dependent architectures like FNO and thereby enabling high-fidelity generation on complex and irregular domains. To complement this architectural advance, we validate the Sliced Wasserstein Distance (SWD) and Maximum Mean Discrepancy (MMD) as general-purpose metrics for fair and standardized model comparison. Our comprehensive experiments confirm the success of this approach: MINO variants achieve state-of-the-art performance across a diverse suite of benchmarks while being substantially more computationally efficient than strong competitors.

Additionally, the MINO framework helps bridge the gap between math-driven operator learning and modern deep learning practice, unlocking the potential to apply high-fidelity generative modeling to a broader range of complex scientific problems. We believe our contributions will foster more rigorous and rapid advancement in the field of functional generative modeling. Python code is available at `https://github.com/yzshi5/MINO`

## Acknowledgments

The authors would like to thank the TMLR reviewers and the action editor for their constructive feedback, and in particular reviewer fkny for the valuable suggestion regarding Table 8. This material is based upon work supported by the U.S. Department of Energy, Office of Science, Office of Advanced Scientific Computing Research, Science Foundations for Energy Earthshot under Award Number DE-SC0024705. ZER is supported by a fellowship from the David and Lucile Packard Foundation.

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

## A  Experimental setup

### A.1  Datasets description

We curated a benchmark of six challenging functional datasets to evaluate performance across different domain types, as summarized in Table 1.

**Navier-Stokes.** This dataset contains solutions to the 2D Navier-Stokes equations on a torus at a resolution of $64 \times 64$. Following the pre-processing of previous work (Kerrigan et al., 2023b; Shi et al., 2025), we use 30,000 samples for training and 5,000 for testing, drawn from the original dataset introduced in (Li et al., 2021).

**Shallow Water.** This dataset contains solutions to the shallow-water equations for a 2D radial dam-break scenario on a square domain, from PDEBench (Takamoto et al., 2022). Each of the 1,000 simulations has 1,000 time steps at $128 \times 128$ resolution; we downsample spatially to $64 \times 64$ for efficiency and treat each time step as an independent snapshot. We randomly select 30,000 snapshots for training and 5,000 for testing

**Darcy Flow.** This dataset contains steady-state solutions of 2D Darcy Flow over the unit square, obtained directly from the PDEBench benchmark (Takamoto et al., 2022). We downsample the original $128 \times 128$ resolution to $64 \times 64$. The dataset contains 10,000 samples and we split it into 8,000 samples for training and 2,000 for testing.

**Cylinder Flow.** We use the Cylinder Flow dataset of Han et al. (2022), which describes flow past a cylinder on a fixed mesh of 1,699 nodes. Each sample is a 3-channel function (x-velocity, y-velocity, pressure). From 101 simulations $\times$ 400 time steps, we ignore temporal order and treat each time step as an independent sample, randomly selecting 30,000 training and 5,000 testing samples.

**Mesh GP.** This is a synthetic dataset generated on a fixed irregular mesh of 3,727 nodes provided by (Zhao et al., 2022). We generate function samples from a Gaussian Process (GP) with a Matérn kernel (length scale = 0.4, smoothness factor = 1.5) given the domain, creating a training set of 30,000 samples and a test set of 5,000 samples.

**Global Climate.** We use the real-world global climate dataset from (Dupont et al., 2022), which contains global temperature measurements over the last 40 years. Each data sample is a function defined on a grid of $46 \times 90$ evenly spaced latitudes and longitudes. Following the previous pipeline (Dupont et al., 2022), we convert the latitude-longitude pairs to Euclidean coordinates ($\mathbb{R}^3$) before passing them to the models. The dataset contains 9,676 training samples and 2,420 test samples.

### A.2  Baselines description

For all baseline models, we adopt their official implementations to ensure reproducibility; the corresponding code repositories are linked in the referenced papers. The key initialization hyperparameters for the baselines : GINO (Li et al., 2023), UPT (Alkin et al., 2024), Transolver (Wu et al., 2024), LNO (Wang & Wang, 2024), and FNO (Li et al., 2021)—are detailed in Table 6. Note that a slightly different configuration was used for the Global Climate experiment. For a complete understanding of all arguments, we refer the reader to the official repository for each respective model (github link provided in all referred papers)

### A.3  Hyperparameters for MINO

For the decoder, the single cross-attention block is followed by a multi-head self-attention (MHSA) block. Although this MHSA block is optional, we found that its inclusion slightly improves performance for generation task. To maintain the overall linear time complexity of MINO, the MHSA module can be replaced by any self-attention variant with linear complexity (like transolver layer).

**MINO-T**. The GNO maps input functions to a latent representation on a $16 \times 16$ grid of query points, defined over the $[0, 1]^2$ domain. We set the GNO search radius to 0.07, the latent dimension $L_{\text{dim}}$ to 256, and the number of attention heads to 4. The encoder consists of $M_1 = 5$ cross-attention blocks. For the Global Climate dataset, which is defined on a spherical manifold $\mathbb{S}^2$, we adjust the latent query positions to a $32 \times 16$ spherical grid and increase the GNO radius to 0.2 to account for the different coordinate system; other hyperparameters remain unchanged. The total parameter count for MINO-T is 21.5 M.

Table 6: Hyperparameter settings for all baseline models.

| Hyperparameter | GINO | UPT | Transolver | LNO | FNO |
|---|---|---|---|---|---|
| Parameters | 19.7 M | 19.6 M | 15.0 M | 22.2 M | 20.6 M |
| GNN radius | - | 0.07 | - | - | - |
| GNO radius | 0.06 | - | - | - | - |
| Width/Dim. | 128 | 192 | 512 | 512 | 128 |
| Blocks/Layers | 4 | 12 | 10 | $8 \times 4$ | 4 |
| Attn. Heads | - | 3 | 6 | 8 | - |
| Latent Grid | $32 \times 32$ | - | - | - | - |
| Latent Tokens | - | 256 | - | 256 | - |
| Fourier Modes | 16 | - | - | - | 24 |
| FNO Channels | 180 | - | - | - | 128 |
| Slice Number | - | - | 24 | - | - |

**MINO-U.** The GNO maps input functions to a latent representation on a $16 \times 16$ grid of query points, defined over the $[0,1]^2$ domain. We set the GNO search radius to 0.07, the latent dimension $L_{\text{dim}}$ to 256, and the number of attention heads to 4. The encoder consists of $M_1 = 2$ cross-attention blocks. For its latent-space processor, we adopt a Diffusion U-Net architecture from the `torchcfm` library (Tong et al., 2024), which operates on the $[16, 16]$ latent tensor with 64 channels, 1 residual block, and 4 attention heads for the processor. Same setting of MINO-T on the Global Climate dataset. The total parameter count for MINO-U is 19.2 M.

### A.4 Details for training and inference

**Reference Measures.** We choose $\mu_0$ as a Gaussian measure characterized by a Gaussian Process (GP) with a Matérn kernel. Unless otherwise specified, the function domain is $[0,1]^2$ (or subset of it), and we use a kernel length scale of 0.01 and a smoothness parameter of 0.5. For the Global Climate dataset, the domain is a spherical manifold represented as a subset of $[-1,1]^3$, and the kernel length scale is adjusted to 0.05. Furthermore, for the GP on the sphere, we explored two distance metrics for the kernel. The first uses the chordal distance (the Euclidean distance in the $\mathbb{R}^3$ embedding space), while the second employs the geodesic distance on $\mathbb{S}^2$, for which we leverage the `GeometricKernels` library (Mostowsky et al., 2024). Although both approaches yield a valid GP, the results presented in this paper are based on the chordal distance implementation. We selected this method for two primary reasons: its superior computational speed and to ensure consistency with other benchmarks. To facilitate further exploration, we will release the code repository with both implementations. The variance is fixed at 1 for all GPs.

**Training Details.** We train all models for 300 epochs using the AdamW optimizer with an initial learning rate of 1e-4. We employ a step learning rate scheduler that decays the learning rate by a gamma of 0.8 every 25 epochs. The default batch size is 96. However, Transolver consumes significantly more GPU memory than other models. To accommodate it on a single NVIDIA RTX A6000 Ada GPU (48 GB memory), we reduced its batch size to 48 and, to maintain a comparable training iteration (duration), limited its training to 200 epochs. Despite these adjustments, Transolver still required approximately 1.76x more total GPU-hours than MINO-T and MINO-U, respectively. The only exception was the Global Climate experiment, where we explicitly matched the total GPU computation time of MINO variants to that of Transolver by extending their training epochs. All experiments reported in Table 4 and Table 3 were conducted three times, and we report the best performance (among the three) achieved for each model. To further strengthen our comparative analysis, Appendix G details an experiment with matched settings (parameter counts, batch size, epochs) that reveals MINO's significant speedup and superior performance over Transolver.

**Inference Details.** To evaluate the models, we generate the same number of samples as contained in the test set for each dataset shown in Table 2. All samples are generated by solving the learned ODE numerically using the `dopri5` solver from the `torchdiffeq` library (Chen et al., 2018), with an error tolerance set to 1e-5 for all experiments.

# B   SWD and MMD as general metrics for functional generation tasks

Unlike image generation, where established metrics like the Fréchet Inception Distance (FID) can leverage a well pretrained models to compare distributions in a latent space, the field of functional generative models lacks such standardized evaluation protocols. Functional data covers a wide range of modalities, often without a common pretrained model, what's more, the learnt objective is probability measure not probability distribution, which requires the metrics should be consistent regardless of discretization of function sample. This makes fair and direct comparison between models challenging, as previous studies frequently rely on dataset-specific metrics. To address this gap, our work proposes and validates two general metrics that directly estimate the distance between a learned probability measure, $\nu$, and a target measure, $\mu$, using samples drawn from each.

We are given two sets of i.i.d. function samples, $X = \{x^{(j)}\}_{j=1}^N$ drawn from $\nu$ and $Y = \{y^{(j)}\}_{j=1}^N$ drawn from $\mu$. Each function is observed at the same $N_{in}$ locations. To compute the distance, each function sample is first flattened into a single vector representation.

**Sliced Wasserstein Distance (SWD)**. In practice, we estimate the distance between two probability measure,$\mu$ and $\nu$, using the discretized function samples in the datasets $X$ and $Y$. To make the metric shown in Eq. 9 computable, we need to derive the discretized version of it, which is achieved by choosing the base measure $\gamma_s$ to be the Haar measure (uniform surface measure) on the sphere $\mathbb{R}^{d-1}$ ($d$ is the discreziation of functions and $d = N_{in}$ if co-domain is 1 in our case) and replace the integral and normalization with an expectation with respect to the probability measure defined by normalizing $\gamma_s$. Finally, we can get a discretized version of Eq. 9:

$$\mathrm{SWD}_p(\mu, \nu) = \left( \mathop{\mathbb{E}}_{\theta \sim \mathcal{U}(\mathbb{S}^{d-1})} \left[ \mathcal{W}_p^p(\theta_\# \mu, \theta_\# \nu) \right] \right)^{\frac{1}{p}} \tag{11}$$

where $\theta$ is a random direction on the unit sphere $\mathbb{S}^{d-1}$. SWD converges to the exact p-Wasserstein distance for two probability measures as the number of projections goes to infinity. In our experiments, we use the official implementation from the `POT` library (Flamary et al., 2021). We use the Sliced Wasserstein-2 distance ($p = 2$) (Bonneel et al., 2015), the time complexity for SWD is $\mathcal{O}(L \cdot N \cdot N_{in} + L \cdot N \log N)$, where $L$ is the number of projections.

**Maximum Mean Discrepancy (MMD).** For MMD (Gretton et al., 2012), the time complexity is $\mathcal{O}(N^2 \cdot N_{in})$ we use the Gaussian RBF kernel $k(\cdot, \cdot)$. Given the sample sets $X$ and $Y$, we compute the squared MMD using the standard unbiased U-statistic estimator:

$$\widehat{\mathrm{MMD}}_u^2(X, Y) = \frac{1}{N(N-1)} \sum_{i \neq j} k(x^{(i)}, x^{(j)}) \ + \ \frac{1}{N(N-1)} \sum_{i \neq j} k(y^{(i)}, y^{(j)}) \ - \ \frac{2}{N^2} \sum_{i=1}^N \sum_{j=1}^N k(x^{(i)}, y^{(j)}) \tag{12}$$

A practical consideration for the SWD is that it relies on a Monte Carlo approximation over random projections. This can introduce variance; calculating the SWD multiple times on the same two test datasets may yield slightly different results. To ensure our evaluations are stable and meaningful, we must reduce this variance to a negligible level. We propose a simple procedure we term **averaged-SWD**: instead of a single SWD estimation, we perform $n_{\mathrm{run}}$ independent estimations via Eq 11(each with a large number of projections, $L = 256$ in our case) and report the mean, we would expect averaged-SWD has very small variance introduced by the Monte Carlo approximation.

To validate this approach, we created two distinct Gaussian measures, characterized by GP, $\mathrm{GP}_1$ (length scale=0.3, smoothness=1.5) and $\mathrm{GP}_2$ (length scale=0.01, smoothness=0.5), on a $64 \times 64$ regular grid. We drew 5,000 GP samples from each measure to create $X, Y$ and then calculated the averaged-SWD between them for varying $n_{\mathrm{run}}$ and repeated this entire process 20 times to report the mean and standard deviation of the averaged-SWD. As shown in Table 7, when $n_{\mathrm{run}} = 10$, the variance of the averaged-SWD is sufficiently small. We therefore adopt $n_{\mathrm{run}} = 10$ for all SWD computations in our experiments. In contrast, MMD is deterministic for a given kernel and does not exhibit this variance.

**Metric Consistency Across Discretizations.** A crucial property of a metric for functional data is its consistency with respect to the discretization of the function domain. The computed distance between two measures should be consistent regardless of the discretization for functions samples in two function datasets drawn from the two probability measure. That is, if we draw new function samples $X_\dagger$ and $Y_\dagger$ from the same measures $\mu$ and $\nu$ but observe them on a different set of locations, we expect $\mathrm{MMD}(X_\dagger, Y_\dagger) \approx \mathrm{MMD}(X, Y)$ and $\mathrm{SWD}(X_\dagger, Y_\dagger) \approx \mathrm{SWD}(X, Y)$.

To verify this property, we calculated SWD and MMD in three representative scenarios while varying the number of observation points by randomly sub-sampling the full discretization ($N_{in}$). The scenarios are: (i). $\mathrm{GP}_1$ vs. $\mathrm{GP}_2$. (ii). $\mathrm{GP}_1$ vs. Navier-Stokes (test dataset) .(iii) $\mathrm{GP}_1$ vs. Cylinder Flow (test dataset). For each case, we will unify the observed position (discretization) of function samples

As shown in Fig. 4, both MMD and SWD remain highly consistent across different sub-sampling ratios of the observation points. This analysis demonstrates that SWD and MMD are not only theoretically sound but also empirically robust and sample-efficient evaluation metrics for functional generative models.

Table 7: Variance analysis of the averaged-SWD metric. We report the mean and standard deviation of the averaged-SWD, calculated over 20 trials, for different numbers of averaging runs ($n_{\mathrm{run}}$).

| $n_{run}$ | 5 | 10 | 20 | 40 | 60 |
|---|---|---|---|---|---|
| SWD (mean $\pm$ std) | $0.2482 \pm 0.0052$ | $0.2485 \pm 0.0038$ | $0.2494 \pm 0.0029$ | $0.2497 \pm 0.0017$ | $0.2498 \pm 0.0014$ |

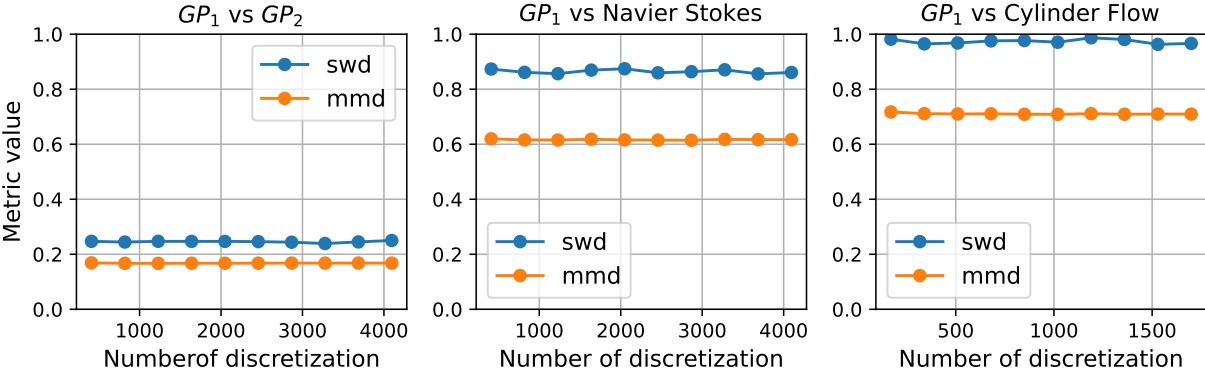

Figure 4: Consistency of SWD and MMD with respect to the number of discretization points. The metrics remain stable as the number of observation points is varied for three comparison scenarios: two synthetic GPs (left), a GP vs. a regular grid dataset (center), and a GP vs. an irregular mesh dataset (right).

## C   Connection and Difference to Prior Works

### C.1   Functional Generative Models vs. Surrogate PDE Solvers

While both functional generative models and surrogate PDE solvers utilize neural operators as their backbone architecture, the demands placed on the operator in these two settings are fundamentally different. A surrogate PDE solver typically learns a one-step mapping between function states, such as mapping initial conditions to a final solution. In contrast, a functional diffusion or flow model must learn to parameterize a continuous transformation between two probability measures, and also requires many steps evaluation during inference. This contrast is further highlighted by the signal frequencies involved: surrogate solvers often map between smooth, low-frequency functions (Li et al., 2023), while functional generative models must transform high-frequency noise into structured data samples. Consequently, designing a neural operator for generative tasks is significantly more demanding, as the learning task itself is more challenging. There are some finite-dimensional generative models (Havrilla et al., 2024; Wang et al., 2024) can also achieve zero-shot generation at arbitrary resolutions. However, these approaches do not explicitly learn a probability measure

over the function space and the zero-shot generation is similar to the linear or bicubic interpolation, which distinguishes them from the functional generative paradigm discussed here.

## C.2 Architectural Comparison to Other Neural Operators

The MINO architecture builds upon insights from previous work, including GINO and UPT (Li et al., 2023; Alkin et al., 2024), but introduces several key modifications to address their limitations in the context of functional generative tasks.

**Comparison with GINO.** GINO adopts a GNO encoder and decoder with an FNO processor. However, since a GNO acts as a low-pass filter (Laplacian smoothing) (Li et al., 2018), which creates an information bottleneck for decoding. The GNO decoder receives only the low-frequency output from the FNO processor, making it structurally difficult to reconstruct the high-frequency details essential for high-fidelity generative modeling, which is supported by our experiments (see Appendix E).

**Comparison with UPT.** The UPT model uses a GNN for super-node pooling, a choice we identify as suboptimal for operator learning for three primary reasons. First, as the input discretization becomes finer, the GNN converges to a pointwise operator, which decouples it from the discretization convergence property required of a neural operator. Second, GNN can also be viewed as a low pass filter in the generative tasks, which has the same issue of GNO. Third, UPT relies on random subsampling to define super-nodes is ill-suited for adaptive meshes common in PDE tasks, where important regions with low grid density may be undersampled. For instance, on a dense mesh of the Earth, random subsampling would likely over-concentrate super-nodes in the polar regions that of less interest. Furthermore, UPT's decoder uses a single-layer, perceiver-style (Jaegle et al., 2022) cross-attention where the queries are derived only from the output positions, which struggles to reconstruct the high-frequency information (Dong et al., 2021). We found that for generative tasks, it is crucial to use a combination of the input function and its position as the query to the cross attention decoder and to use multiple attention layers as shown in the ablation study (Appendix E). In Table 8, we present a detailed architectural comparison between MINO and GINO/UPT.

**Comparison with Transolver.** Key differences lie in the receptive field and latent-space design of MINO. Transolver employs a purely global receptive field, whereas our architecture integrates both local and global information processing. In addition, MINO follows the modern, highly scalable, and efficient DiT paradigm by replacing the grid-dependent "patchify" operation with a domain-agnostic GNO encoder. Finally, because the slicing and de-slicing operations in Transolver act on every point, and these operations happen in every Transolver layer, which can be inefficient. In contrast, MINO performs encoding and decoding only once and can be viewed as a reduced-order operator; its decoder is shallow, consisting of one cross-attention block followed by a single self-attention block. As shown in our experiments, MINO is markedly more efficient than Transolver in both training and inference.

## C.3 Designing neural operators with general deep learning architectures

The composition of multiple neural operators remains a valid neural operator (Wang et al., 2025). Previous works such as GINO and UPT adopt the standard encoder–processor–decoder paradigm. However, directly replacing the processor with a generic neural network is sub-optimal: it forces the decoder to rely solely on a fixed-size latent representation, whereas the desired output lives in an infinite-dimensional function space. To overcome this limitation, we design the decoder via cross attention to take both the latent representation and the original input function as inputs, thereby circumventing the bottleneck imposed by a fixed latent shape.

# D Additional Results

In this part, we present additional qualitative results to further demonstrate the performance of MINO compared to baseline models. As shown in Figure 5, MINO generates realistic samples for the Navier-Stokes benchmark that closely match the ground truth. In contrast, both GINO and UPT capture only the general, low-frequency trends of the solution, failing to reconstruct the high-frequency information essential for high-fidelity generation. This difficulty arises because generative models must learn a velocity field with correct high-frequency components to successfully transform high-frequency Gaussian noise into structured data samples. We provide further qualitative results for MINO on the Darcy Flow, Shallow Water, and

Table 8: Architectural Comparison: MINO vs. GINO / UPT

| Feature | MINO (Mesh-Informed Neural Operator) | GINO / UPT Frameworks |
|---|---|---|
| **Primary Goal** | Probabilistic **Functional Generative Modeling** (learning distributions via flow matching). | Deterministic **Physics Simulation**: **GINO**—solving PDEs on varying geometries; **UPT**—scalable temporal simulation (latent rollouts). |
| **Encoder Strategy** | **Learned Resampling onto a Grid**: a Graph Neural Operator (GNO) projects from $N_{in}$ irregular points to a fixed, regular $N_{node}$ grid. | **GINO:** GNO maps points to a regular grid and augments them with geometric features (signed-distance function). **UPT:** a GNN pools information from $N_{in}$ points onto a smaller, fixed set of $S$ "supernodes." |
| **Latent Representation** | **Structured Feature Map (Grid)**: a tensor of shape $(H \times W \times C)$, compatible to CNNs. | **GINO:** structured feature map (grid). **UPT:** unstructured set of vectors (**tokens**) of shape $(N_{tokens} \times D_{hidden})$. |
| **Core Processor** | SOTA diffusion-style U-Net (or Transformer). | **GINO:** Fourier Neural Operator (FNO) for efficient global processing on the latent grid. **UPT:** Transformer (Approximator) for modeling temporal transitions between latent tokens. |
| **Decoder Mechanism** | Cross-Attention **with Input Bypass: Query (Q)** comes from the original input function $f_t$ and its positions; **Key/Value (KV)** come from the processed latent state. Prevents an information bottleneck and preserves high-frequency details. | **No Input Bypass (Information Bottleneck): GINO** projects directly from latent grid to query points. **UPT** uses Perceiver-style cross-attention where **Q** is derived from *output positions only*. All information must be reconstructed solely from the compressed latent state. |
| **Uncertainty** | **Inherent & Computationally Cheap:** being generative, the model learns a distribution; Monte-Carlo dropout can provide epistemic uncertainty with a **single** trained network. | **Computationally Expensive:** uncertainty is not a primary target; estimating it usually requires training an **ensemble of models** (multiple runs with different seeds). |

Mesh-GP benchmarks in Figures 6, 7, and 8, respectively, which visually confirm the effectiveness of our approach.

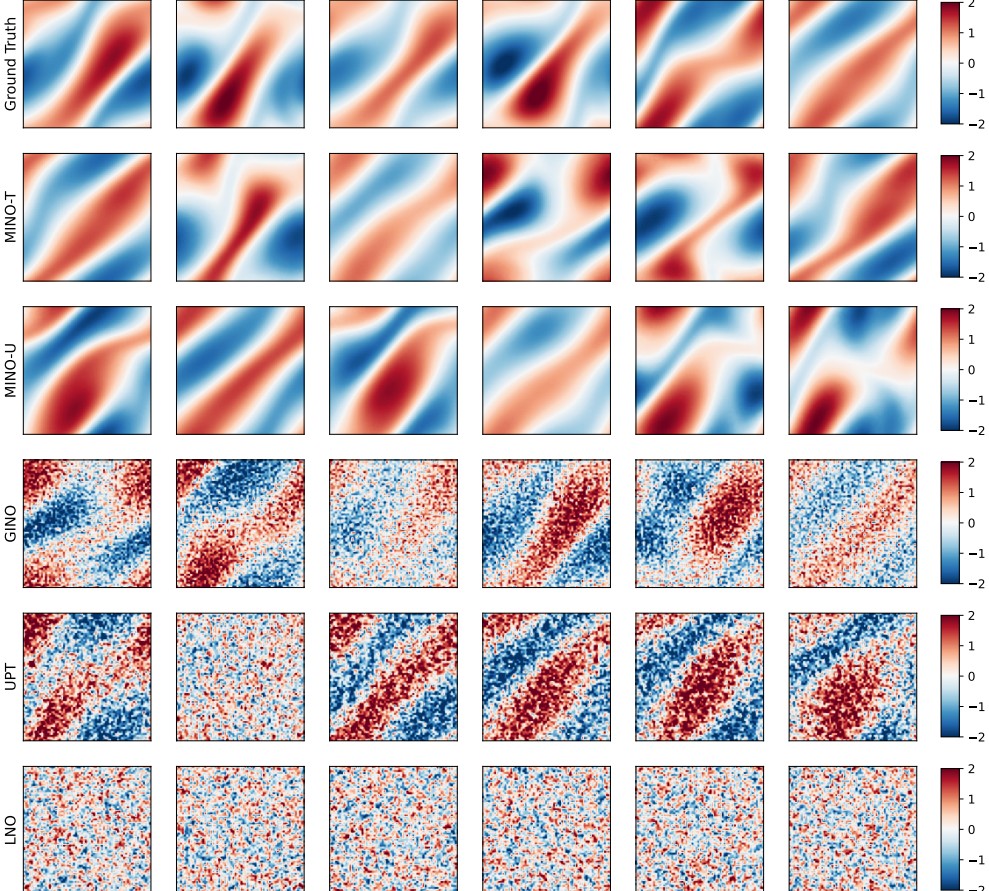

Figure 5: Generation of Navier-Stokes samples under different baselines

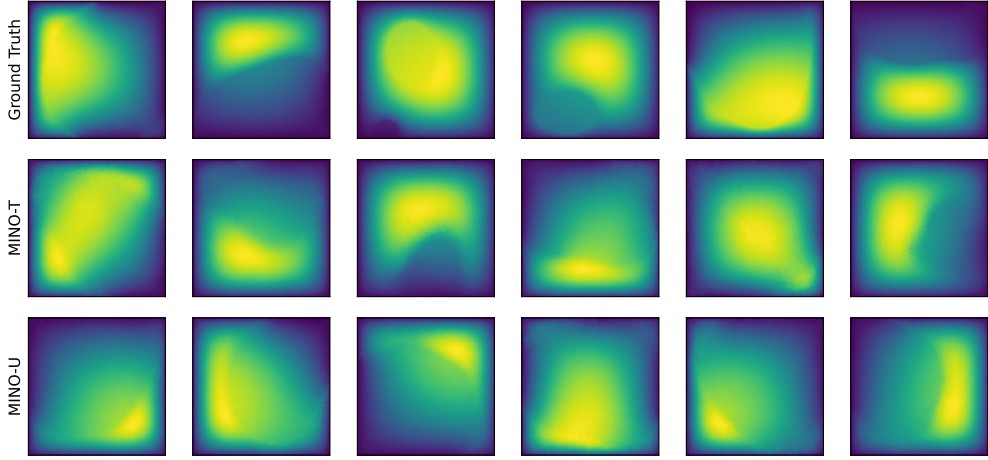

Figure 6: Generation of Darcy-Flow samples

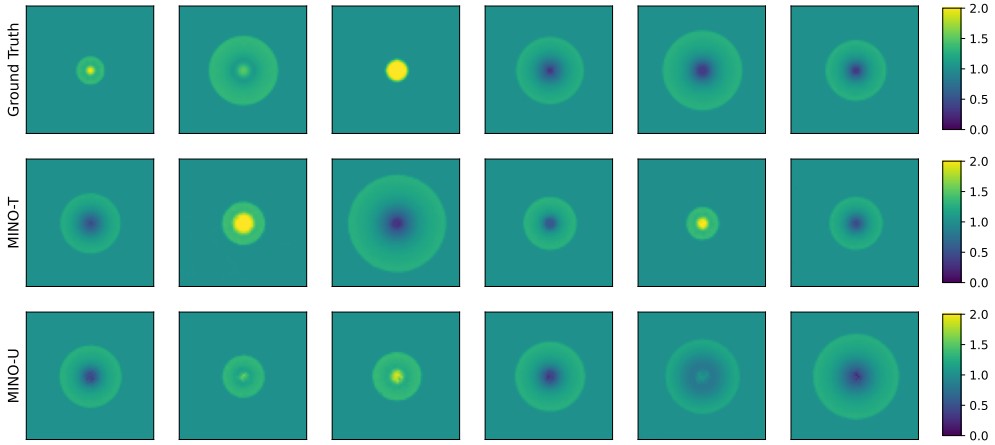

Figure 7: Generation of Shallow Water samples

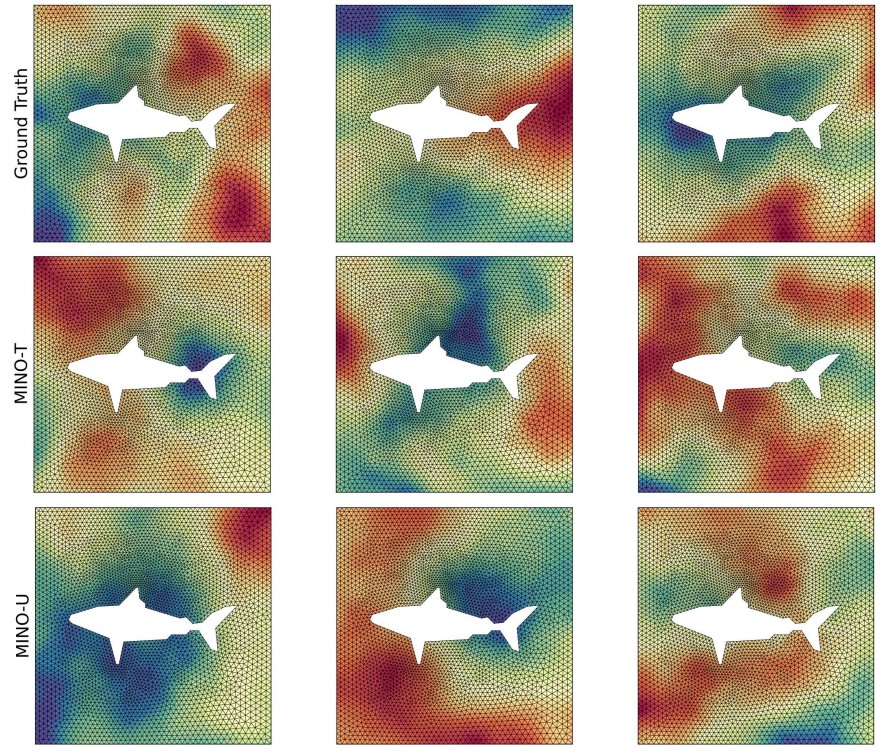

Figure 8: Generation of Mesh GP samples

# E    Ablation Study

We conduct an ablation study to validate the effectiveness of the specific cross-attention mechanisms used in the encoder and decoder of our MINO architecture. We take the standard MINO-T as our baseline and evaluate its performance against two architectural variants:

**Self-Attention Encoder:** We replace the cross-attention mechanism in the encoder with a standard self-attention mechanism. In this setup, the Key-Value (KV) pair for each attention block is derived from the output of the previous block, rather than being fixed as the initial latent representation from the GNO.

**Position-Only Decoder:** We modify the decoder's query to be derived only from the function's observation positions, similar to the decoder design in prior works like UPT, GINO. This removes the explicit dependency on the input function's values in the query.

The results of this study, presented in Table 12, confirm that our proposed design choices are critical for achieving high performance. The standard MINO-T consistently and significantly outperforms both ablated variants across all datasets and metrics. The performance degradation is most severe in the *Position-Only Decoder* variant. By relying only on positional queries, the decoder struggles to reconstruct fine-grained details. This result highlights the importance of conditioning the decoder's queries on the input function's values for high-fidelity generative tasks. The *Self-Attention Encoder* variant also performs worse than our standard model, which suggests that using the initial latent representation as a fixed context for iterative refinement in the encoder is a more effective strategy than a standard self-attention approach.

Table 9: Ablation study on key architectural components of MINO across four benchmarks. We compare our standard MINO-T against two variants: one replacing the encoder's cross-attention with self-attention, and another using a simpler position-only decoder query similar to prior work like UPT. Lower SWD/MMD scores are better; best results are in bold

| Dataset → | Navier Stokes | | Shallow Water | | Darcy Flow | | Cylinder Flow | |
|---|---|---|---|---|---|---|---|---|
| Variants ↓ Metric → | SWD | MMD | SWD | MMD | SWD | MMD | SWD | MMD |
| Standard MINO-T | $\mathbf{4.0 \cdot 10^{-2}}$ | $\mathbf{3.6 \cdot 10^{-2}}$ | $\mathbf{9.8 \cdot 10^{-3}}$ | $\mathbf{8.7 \cdot 10^{-3}}$ | $\mathbf{8.9 \cdot 10^{-2}}$ | $\mathbf{3.4 \cdot 10^{-2}}$ | $\mathbf{2.9 \cdot 10^{-2}}$ | $\mathbf{2.6 \cdot 10^{-2}}$ |
| Self-Attention Encoder | $5.2 \cdot 10^{-2}$ | $5.1 \cdot 10^{-2}$ | $1.7 \cdot 10^{-2}$ | $1.7 \cdot 10^{-2}$ | $2.0 \cdot 10^{-1}$ | $7.6 \cdot 10^{-2}$ | $3.7 \cdot 10^{-2}$ | $3.5 \cdot 10^{-2}$ |
| Position-Only Decoder | $5.7 \cdot 10^{-1}$ | $4.5 \cdot 10^{-1}$ | $8.4 \cdot 10^{-1}$ | $5.7 \cdot 10^{-1}$ | $8.3 \cdot 10^{-1}$ | $4.4 \cdot 10^{-1}$ | $7.2 \cdot 10^{-1}$ | $5.3 \cdot 10^{-1}$ |

# F    Empirical Analysis of Scaling and Stability

To evaluate performance consistency, we train both MINO-U and MINO-T with eight different random seeds on the Navier-Stokes and Cylinder-Flow datasets. The results, summarized in Table 10, show that MINO-U consistently outperforms MINO-T across both datasets and metrics. We also note that performance variance across seeds is slightly higher on the Navier-Stokes task compared to the Cylinder-Flow task.

We next study data-scaling on Navier–Stokes benchmark by training on progressively larger subsets of the full 30 k-sample dataset and evaluating on the same test split. Table 11 shows steady performance gains as the training set grows, with improvements tapering off once roughly 18 k samples are reached.

Table 10: Performance comparison over eight random seeds. We report the mean ± standard deviation (in the parentheses) for each metric. Lower is better.

| Dataset → | Navier Stokes | | Cylinder Flow | |
|---|---|---|---|---|
| Model ↓ Metric → | SWD | MMD | SWD | MMD |
| MINO-T | $3.7 \cdot 10^{-2}$ $(6.8 \cdot 10^{-3})$ | $3.1 \cdot 10^{-2}$ $(6.0 \cdot 10^{-3})$ | $2.9 \cdot 10^{-2}$ $(2.4 \cdot 10^{-3})$ | $2.6 \cdot 10^{-2}$ $(3.0 \cdot 10^{-3})$ |
| MINO-U | $3.3 \cdot 10^{-2}$ $(6.3 \cdot 10^{-3})$ | $2.7 \cdot 10^{-2}$ $(7.5 \cdot 10^{-3})$ | $2.1 \cdot 10^{-2}$ $(2.0 \cdot 10^{-3})$ | $1.9 \cdot 10^{-2}$ $(2.8 \cdot 10^{-3})$ |

Table 11: Performance of MINO-U when trained on a subset of the Navier-Stokes dataset.

| Metrics ↓ Samples → | 6,000 | 12,000 | 18,000 | 24,000 | 30,000 |
|---|---|---|---|---|---|
| SWD | $4.3 \cdot 10^{-2}$ | $3.9 \cdot 10^{-2}$ | $2.8 \cdot 10^{-2}$ | $3.0 \cdot 10^{-2}$ | $2.8 \cdot 10^{-2}$ |
| MMD | $4.0 \cdot 10^{-2}$ | $3.2 \cdot 10^{-2}$ | $2.2 \cdot 10^{-2}$ | $3.1 \cdot 10^{-2}$ | $1.9 \cdot 10^{-2}$ |

## G   Comparison with Transolver under Identical Settings

To ensure a fair comparison with Transolver, we reduce the latent dimension of MINO from 256 to 192, and use the same training configuration (200 epochs, batch size 48). This results in smaller models—MINO-U (S) (13.5M parameters) and MINO-T (S) (12.1M)—compared to Transolver (15.0M). Notably, our models are significantly faster, achieving a 3.1x speedup in training and a 3.5x speedup in inference while maintaining superior performances.

Table 12: Performance comparison with Transolver under identical settings. Best Performance in bold

| Dataset → | Navier Stokes | | Mesh GP | | Cylinder Flow | |
|---|---|---|---|---|---|---|
| Model ↓ Metric → | SWD | MMD | SWD | MMD | SWD | MMD |
| Transolver | $5.3 \cdot 10^{-2}$ | $5.1 \cdot 10^{-2}$ | $9.1 \cdot 10^{-2}$ | $4.6 \cdot 10^{-2}$ | $2.5 \cdot 10^{-2}$ | $2.4 \cdot 10^{-2}$ |
| MINO-T (S) | $4.4 \cdot 10^{-2}$ | $4.0 \cdot 10^{-2}$ | $7.9 \cdot 10^{-2}$ | $\mathbf{2.8 \cdot 10^{-2}}$ | $2.7 \cdot 10^{-2}$ | $2.6 \cdot 10^{-2}$ |
| MINO-U (S) | $\mathbf{2.9 \cdot 10^{-2}}$ | $\mathbf{2.5 \cdot 10^{-2}}$ | $\mathbf{6.1 \cdot 10^{-2}}$ | $3.0 \cdot 10^{-2}$ | $\mathbf{2.5 \cdot 10^{-2}}$ | $\mathbf{2.3 \cdot 10^{-2}}$ |

