# OpenReview forum: "Mesh-Informed Neural Operator : A Transformer Generative Approach"
_TMLR — Accepted by TMLR_

### Review · Reviewer_fkny · 2025-07-22

**Summary Of Contributions:**

The paper presents MINO (mesh informed neural operator) an improvment of GINO (geometry informed neural operator).

The key differences are

- different encoder/decoder (incorporating a skip connection from input to decoder and adding cross attention to both en- and decoder)
- switch in training paradigm (flow matching for learning distributions of operators vs. deterministic learning of PDEs/unrolled PDEs)
- leveraging the optimizations and training tricks of SotA image diffusion models with a UNET/transformer latent processor

which are evaluated on a set of regular and irregular grid datasets, showing improvements across the board

**Audience:**

Yes

**Broader Impact Concerns:**

Not applicable/sufficient

**Claims And Evidence:**

Yes

**Requested Changes:**

# critical

-  perform bootstrapping + a suitable statistical significance test on the test set as a cheap way to add statistical significance (w.r.t to the dataset)


# improvement
## clarify what is mean by "systematic evaluation of modern architectures for functional generation"

Did prior work not to modern architectures? not systematic? not functional?  (this one is a nit pick, but one of my triggers)

## train multiple seeds and report mean and std
In the same experiment, reduce parameter count of MINO to matchthe count of a transolver such that transolver can fit into the GPU memory (consider checkpointing, mixed precision training etc.) to make a fair comparison (could be worth doing a mini-scaling study on a subset/coreset of the datasets as well, but that's already a bonus idea)

## Concise delta w.r.t to GINO and UPT

I struggled a bit to find the concrete distinctions between GINO and this paper,the below table represents my best attempt after a close reading of the paper + the references. I suggest to add something like this to the appendix or main body

**Architectural Comparison: MINO vs. GINO / UPT**

| Feature | **MINO (Mesh-Informed Neural Operator)** | **GINO / UPT Frameworks** |
| :--- | :--- | :--- |
| **Primary Goal** | Probabilistic **Functional Generative Modeling** (learning distributions via flow matching). | Deterministic **Physical Simulation**:  **GINO:** Solving PDEs on varying geometries.  **UPT:** Scalable temporal simulation (latent rollouts). |
| **Encoder Strategy** | **Learned Resampling onto a Grid:** GNO projects from $N_{in}$ irregular points to a fixed, regular $N_{node}$ grid. | **GINO:** GNO maps points to a regular grid, augmented with geometric features (SDF).  **UPT:** GNN (not GNO) **pools** information from $N_{in}$ points onto a smaller, fixed set of $S$ "supernodes". |
| **Latent Representation**| **Structured Feature Map** (`Grid`): A tensor of shape ($H \times W \times C$), compatible with CNNs. |  **GINO:** Structured feature map (`Grid`).  **UPT:** Unstructured set of vectors (`Tokens`) of shape ($N_{tokens} \times D_{hidden}$). |
| **Core Processor**| SotA Diffusion-style U-Net (or Transformer)   | **GINO:** Fourier Neural Operator (FNO) for efficient, global processing on the latent grid. **UPT:** Transformer (`Approximator`) for modeling temporal state transitions between latent tokens. |
| **Decoder Mechanism** | Cross-Attention **with Input Bypass:**  **Query ($Q$):** Derived from **original input function $f_t$** and its positions. • **Key/Value ($KV$):** Derived from the processed latent state. This prevents an information bottleneck, preserving high-frequency details. | **No Input Bypass (Information Bottleneck):** **GINO:** projects from the latent grid to query points.  **UPT:** Perceiver-style Cross-Attention where **Query ($Q$)** is derived from **output positions only**. All information must be reconstructed solely from the compressed latent state. |
| **Uncertainty** | **Inherent & Computationally Cheap:** As a generative model, it learns a distribution. Can leverage **Monte Carlo Dropout** to estimate epistemic uncertainty from a single trained model. | **Computationally Expensive:** Not a primary goal. Requires training a full **ensemble of models** (i.e., multiple runs with different seeds) to estimate model uncertainty. |

**Strengths And Weaknesses:**

Strength

- the approach is will motivated and explained in it's functionality
- very clever remix of GINO & UPT (and I mean this as a confidence) , with modifications that I can understand alleviating the oversmoothing issues of GINO and UPT (it might be worth adding https://arxiv.org/abs/2206.10991 and https://arxiv.org/abs/2103.03404 as references explaining the effects in both GNNs and attention)
- systematic evaluation and ablations

Weaknesses

- single seeds reported without attempts at uncertainty quantifification
- mildly unfair training/evaluation (different batchsize for transolver, different parameter counts)
- while the method is well explained for _understanding_ its _distinction_ could be improved (see below)

---

> ### Author Response · Authors · 2025-08-07
> **Official Comment by Authors**
>
> We appreciate the reviewer’s valuable comments and the positive take on our work. In the following, we address the comments and explain changes made in the main text. These changes in the main text are colored in blue in the revised draft.
>
> **S1. Very clever mix of GINO and UP. It might be worth adding related references explaining the effects in both GNNs and attention.**
>
> Great suggestion. We have now incorporated the mentioned references (Di Giovanni et al, 2023; Dong et al, 2023) into the preliminaries section to better explain the effects in both GNNs and attention mechanisms.
>
> References :
>
> Di Giovanni et al, “Understanding convolution on graphs via energies” 2023
>
> Dong et al, “Attention is Not All You Need: Pure Attention Loses Rank Doubly Exponentially with Depth” 2023
>
>
> **W1 & Improvement 2:  Single seeds reported without attempts at uncertainty quantification, train multiple seeds and report mean and std. Could be worth doing the scaling study on a subset/coreset of the dataset as well.**
>
> We appreciate this constructive feedback. We have now run our experiments with multiple seeds and included a dataset scaling study as suggested. The full results, including mean and standard deviation, are now available in Appendix F of the updated draft
>
>
> **W2. Mildly unfair training/evaluation (different batchsize for transolver, different parameter counts)**
>
> Thank you for this comment. To ensure a fair comparison, we have run new experiments using smaller MINO models, strictly matching the batch sizes, training epochs and parameter counts of the Transolver baseline.
>
> Under this direct comparison, our model outperforms the baseline on key metrics while also demonstrating a 3.1x training speedup and a 3.5x inference speedup. We have added these stronger results to Table 12 in Appendix G and updated the corresponding text.
>
>
>
>
> **W3 & Improvement 3. While the method is well explained for understanding its distinction could be improved (see below)**
>
> We thank the reviewer for this highly constructive feedback. The suggested table provides an outstanding summary of our method's key distinctions.
> Following this recommendation, we have added the table to the manuscript to better contextualize our work relative to GINO and UPT. This comparison is now located in Appendix B (Table 8, Page 19)
>
>
> **Improvement 1. Clarify what is mean by “systematic comparison of modern neural-operator architectures on function-generation problems.”  (this one is a nit pick, but one of my triggers)**
>
> Thank you for this question. While functional generative modeling theories have been proposed (Rahman et al, 2022; Kerrigan et al,  2023; Shi et al, 2025; Lim et al, 2025.), their practical implementations have been confined to older FNO-based architectures on regular grids. In parallel, significant progress has been made on modern neural operators (UPT, Transolver, GINO, etc) for PDE problems on irregular grids.
>
> Our work is the first to systematically investigate if these modern advancements in operator learning can translate to the domain of function generation. We adapt, test, and compare these state-of-the-art architectures on generative tasks, particularly on the irregular grids they were designed for, providing the first head-to-head comparison in this new context.
>
> Reference:
>
> Rahman et al, “ Generative Adversarial Neural Operators”, 2022
>
> Kerrigan et al, “Functional Flow Matching” 2023
>
> Shi et al, “ Stochastic Process Learning via Operator Flow Matching” 2025
>
> Lim et al, “Score-based Diffusion Models in Function Space” 2025

---

### Review · Reviewer_4dSS · 2025-08-28

**Summary Of Contributions:**

In this work, the authors propose a novel architecture Mesh-Informed Neural Operators (MINO) which extents existing Neural Operator frameworks (in particular, Fourier Neural Operators) for functional generative models. MINO does not needed a rectangular grid nor a rectangular domain. MINO is still discretization agnostic, and so allows for the application of Neural Operators to new application areas. The authors also introduce two new functional generative evaluation metrics, one based on the sliced Wasserstein distance the other maximum mean discrepancy, to test MINO against existing architectures. Through a convincing set of experiments, the authors demonstrate that MINO achieves state-of-the-art performance on various benchmarks.

**Audience:**

Yes

**Broader Impact Concerns:**

No broader impact concerns.

**Claims And Evidence:**

Yes

**Requested Changes:**

Typos
 - In Eqn. 3, should the plus sign be concatenation, as in Eqn. 6?
 - The first sentence of Section 4, "baselines on a suite of functional generative benchmarks under OFM (Shi et al., 2025) paradigm" should read "baselines on a suite of functional generative benchmarks under **the** OFM (Shi et al., 2025) paradigm."
 - A complaint about the formatting/standards. Many references are given as references to the arxiv version of the work, rather than the published version of the work. As one (of many) example, Ashish Vaswani et al "Attention Is All You Need" was published in neurips 2017, yet the references in this manuscript is to the arxiv version of the paper. I think the authors should double-check their references to make sure that the references to the works are to the published works when possible.
 - On page 17, the fourth line "That is, if we draw anthor new function samples..." This sentence has a typo.
 - Also page 17, fourth line in the first paragraph of section C.1, "conditions to a final solution ." There is an extra space. It should be "conditions to a final solution."

**Strengths And Weaknesses:**

Strengths:
 - The work is well written and easy to follow.
 - The architecture is presented clearly, and the motivation for the various modules (encoder, processor and decoder) is well-motivated
 - The numerical experiments show that MINO has strong performance across various challenging test problems
 - The proposed functional generative evaluation metrics are likely to be of general interest, independently of MINO, which further strengthens the work.

Weaknesses:
 - The work contains a few minor typos. Please see the requested changes section

---

> ### Author Response · Authors · 2025-08-29
> **Official Comment by Authors**
>
> We appreciate the reviewer’s valuable comments and the positive take on our work. We have carefully revised the manuscript to address the suggested changes.
>
>
>
> **Q1. In Eqn. 3, should the plus sign be concatenation, as in Eqn. 6?**
>
> This is an excellent catch. We have corrected Eq. 3 to use concatenation operation (consistent with Eq. 6). We appreciate the reviewer's careful attention to detail.
>
> **Q2. The first sentence of Section 4, "baselines on a suite of functional generative benchmarks under OFM (Shi et al., 2025) paradigm" should read "baselines on a suite of functional generative benchmarks under the OFM (Shi et al., 2025) paradigm.**
>
> Thank you for pointing this out, we have updated the sentence accordingly.
>
> **Q3. A complaint about the formatting/standards. Many references are given as references to the arxiv version of the work, rather than the published version of the work. e.g, Ashish Vaswani et al "Attention Is All You Need".**
>
> Thank you for this valuable suggestion. We have thoroughly reviewed our bibliography and updated all citations to reference the published versions of the works where available.
>
> **Q4 On page 17, the fourth line "That is, if we draw another new function samples..." This sentence has a typo.**
>
> Thank you for catching this. We have corrected the typo in the revised manuscript.
>
> **Q5. Also page 17, fourth line in the first paragraph of section C.1, "conditions to a final solution ." There is an extra space. It should be "conditions to a final solution."**
>
> We appreciate the detailed proofreading and have removed the extra space.

---

### Review · Reviewer_3VJC · 2025-09-01

**Summary Of Contributions:**

This paper introduces a novel architecture for functional generative models: the Mesh-Informed Neural Operator (MINO). By integrating the strengths of previous methods, MINO effectively captures both local and global receptive fields, handles irregular grids, and learns both low- and high-frequency features. The results show that MINO outperforms existing learning-based methods on both regular and irregular grids.

**Audience:**

Yes

**Broader Impact Concerns:**

No concerns.

**Claims And Evidence:**

Yes

**Requested Changes:**

- In Table 2, are there traditional (non–training-based) methods commonly used for this task? If so, could you include comparisons with such approaches to better position your method?

- Could you also provide experiments on datasets with larger mesh sizes, in order to more convincingly demonstrate the scalability of your approach?

**Strengths And Weaknesses:**

Strengths

- The proposed method effectively integrates the benefits of several prior approaches and demonstrates strong performance across multiple benchmark datasets.

- The paper introduces Sliced Wasserstein Distance (SWD) and Maximum Mean Discrepancy (MMD) as evaluation metrics, which enable fair and comparable assessment across different datasets.

- The writing is clear and well-structured, with a thorough and detailed summary of related work.

Weaknesses

- The comparisons in this paper primarily focus on training-based methods. Including traditional numerical or physics-based baselines would strengthen the evaluation and highlight the practical advantages of the proposed method.

- The Transolver paper reports results on datasets with mesh sizes up to 32,000, whereas this paper only evaluates up to 4,096. This raises concerns about the scalability of the proposed method to significantly larger meshes.

---

> ### Author Response · Authors · 2025-09-01
> **Official Comment by Authors**
>
> We appreciate the reviewer’s constructive feedback and positive evaluation. Below, we provide a point-by-point response to the requested changes.
>
>
> **W1. The comparisons in this paper primarily focus on training-based methods. Including traditional numerical or physics-based baselines would strengthen the evaluation and highlight the practical advantages of the proposed method.**
>
> Thank you for this comment. Our work targets probabilistic functional generative modeling, which is related to—but distinct from—deterministic surrogate PDE solving. We clarify this connection in Appendix C.1 (p. 17). In brief, functional generative models learn a map between probability measures (to produce distributional outputs), whereas surrogate PDE solvers learn deterministic solution operators. Accordingly, it is standard to benchmark surrogate PDE solvers against traditional numerical/physics-based baselines, but not functional generative models. Because the problem formulations and outputs differ (distributional generation vs. single forward solves), direct comparisons to classical numerical solvers are not methodologically aligned.
>
> That said, functional generative models can be specialized for probabilistic PDE solving, an active direction that our framework can support (e.g., Chen et al., 2024; Yao et al., 2025). To keep the evaluation faithful to our primary problem setting, we did not include classical numerical baselines in the main tables.
>
>
> References :
>
> Chen et al “Gradient-Free Generation for Hard-Constrained Systems,” 2024
>
> Yao et al “Guided Diffusion Sampling on Function Spaces with Applications to PDEs”, 2025
>
>
> **W2. The Transolver paper reports results on datasets with mesh sizes up to 32,000, whereas this paper only evaluates up to 4,096. This raises concerns about the scalability of the proposed method to significantly larger meshes.**
>
> Thank you for the question. The 32,000-point mesh case mentioned contains only ~800 training samples (Bonnet et al., 2022; Wu et al., 2024), which is insufficient for learning a probability distribution over functions. As we show empirically in Appendix F, training a robust functional generative model typically requires thousands of samples (≥6,000 in our experiments)—a substantially higher data regime than deterministic surrogates. In practice, collecting high-resolution function datasets with many samples is challenging; accordingly, we emphasize sample count over mesh size for this setting.
>
> Furthermore, our model is discretization-agnostic: it can be trained at a lower resolution and evaluated at higher resolutions without retraining. As shown in Figure 3, we train MINO on Navier–Stokes at 4,096 points and evaluate it at 25,600 points, demonstrating resolution-scalable generation consistent with the learned function-space distribution.
>
> References :
>
> Bonnet et al, “AirfRANS: High fidelity computational fluid dynamics dataset for approximating reynolds-averaged navier–stokes solutions” 2022
>
> Wu et al, “Transolver: A Fast Transformer Solver for PDEs on General Geometries” 2024
>
>
>
> **Requested Change 1. In Table 2, are there traditional (non–training-based) methods commonly used for this task? If so, could you include comparisons with such approaches to better position your method?**
>
> This is a good point. Because our task is probabilistic function generation rather than deterministic PDE solving, classical simulators are not directly comparable in their objective or output. We clarify this scope in Appendix C.1. While extending functional generative models to probabilistic PDE solving is a promising application for our framework (Chen et al., 2024; Yao et al., 2025), it is outside the scope of our direct comparisons.
>
> References:
>
> Chen et al., Gradient-Free Generation for Hard-Constrained Systems, 2024.
>
> Yao et al., Guided Diffusion Sampling on Function Spaces with Applications to PDEs, 2025.
>
>
> **Requested Change 2. Could you also provide experiments on datasets with larger mesh sizes, in order to more convincingly demonstrate the scalability of your approach?**
>
> Constructing high-resolution functional datasets with large sample counts is computationally expensive. For example, the largest dataset cited in Transolver includes only ~800 training samples (Wu et al., 2024), which is below the scale needed for robust distribution learning. We therefore prioritized datasets with sufficient sample size, as this is the dominant factor for functional generative models.
>
> Moreover, because our approach is discretization-agnostic, it supports zero-shot evaluation at higher resolutions. As shown in Figure 3 with the Navier–Stokes dataset, the model is trained at 4,096 points and evaluated at 25,600 points without retraining, confirming its scalability.

---

### Decision · Action_Editor_Tntt · 2025-10-21

**Recommendation:** Accept with minor revision

**Additional Comments:**

This is a clearly written and well-executed paper making several interesting contributions to functional generative modelling. The submitted version was already solid, and the revisions made it substantially better.

Please do the following for the final revision:
1. Make sure you are citing the peer-reviewed version of the paper, if there is one, and include the venue for each paper, as far too many of these are missing. Remove the paper URLs, as they introduce visual clutter without adding much, especially for the axiv publications.
2. Check the number in the top right corner of Table 11. It should probably be 2.8x10^(-2), not 2.8x10^2.
3. Consider acknowledging reviewer fkny for their helpful suggestions, especially the table explaining the differences between MINO amnd GINO/UPT.

**Audience:**

Yes

**Audience Explanation:**

The paper makes several contributions to functional generative modelling, including a domain-agnostic generative backbone, capable of handling irregular grids of observations, and two dataset-independent metrics for evaluating functional models. Thus it will certainly be of interest to a subset of the TMLR audience.

**Claims And Evidence:**

Yes

**Claims Explanation:**

All the reviewers agreed that the claims about the novelty of the contributions and performance of the method were well supported.